# Smart Tourism Technologies, Revisit Intention, and Word-of-Mouth in Emerging and Smart Rural Destinations

**Zabih-Allah Torabi** [1,*], **Mehdi Pourtaheri** [1], **Colin Michael Hall** [2,3,4,5,6], **Ayyoob Sharifi** [7] **and Fazlollah Javidi** [1]

[1] Department of Geography and Rural Planning, Tarbiat Modares University, Tehran 1411713116, Iran; mahdit@modares.ac.ir (M.P.)

[2] Department of Management, Marketing, and Tourism, University of Canterbury, Christchurch 8140, New Zealand; michael.hall@canterbury.ac.nz

[3] Geography Research Unit, University of Oulu, 90014 Oulu, Finland

[4] Department of Service Management and Service Studies, Lund University, 22100 Lund, Sweden

[5] School of Business and Economics, Linnaeus University, 35195 Växjö, Sweden

[6] The College of Hotel & Tourism Management, Kyung Hee University, Seoul 02447, Republic of Korea

[7] The IDEC Institute, Hiroshima University, Hiroshima 734-8553, Japan; sharifi@hiroshima-u.ac.jp

* Correspondence: zabih.torabi@modares.ac.ir

**Abstract:** This study examines the influence of the various attributes of smart tourism technologies (STTs) on tourists' intentions to revisit locations and engage in word-of-mouth (WOM) activities regarding emerging and smart rural tourist destinations in Iran. A sample of 590 tourists who visited these destinations following the COVID-19 pandemic participated in the study. The findings reveal that three attributes of STTs, namely, informativeness, accessibility, and interactivity, positively contribute to tourists' memorable experiences (ME). Furthermore, ME, satisfaction, and the willingness to engage in WOM recommendations are identified as predictors of tourists' intention to revisit rural destinations. The study also reveals that user competence serves as a mediator between the attributes of STTs (informativeness, accessibility, and interactivity) and the tourists' ME. Specifically, tourists with greater skills and knowledge of using STTs tend to have more memorable experiences in these emerging and smart rural destinations. The study discusses both the theoretical and practical implications of these findings.

**Keywords:** smart tourism technologies; memorable experiences; revisit intention; user competence; word-of-mouth; smart rural destinations





## 1. Introduction

During the COVID-19 outbreak, smart rural tourism destinations were established in various parts of the world [1,2]. These destinations attempted to attract those tourists that intended to ensure the least contact with others by offering smart services [3,4]. The COVID-19 pandemic provided the opportunity for rural destinations to develop their information and communication technology (ICT) infrastructure [2,5]. ICT development in these destinations appears to facilitate tourists' exposure to unique experiences, a phenomenon that was not feasible before the pandemic [6]. This suggests that the attributes of smart tourism technologies (STTs) in emerging and smart rural destinations can offer flexible mobility options to tourists, who can modify their routes, accommodation, or type of travel in the face of crises or unforeseen issues [7]. Thus, the attributes of STTs can boost tourists' memorable experiences (ME), satisfaction, and intention to revisit, and may encourage them to offer word-of-mouth (WOM) recommendations [8–10]. In addition, it seems that user competence when utilizing STTs can improve the tourists' ME [11,12]. That is, individuals who are more competent in using STTs can creatively exploit most of the potential of these technologies, hence gaining unique experiences [12–14].

To explore the impact of STTs on tourism destinations, No and Kim [15] classified the attributes of STTs into four categories—accessibility, informativeness, interactivity, and personalization. Easy access to information, high quality of the provided information, enhanced relationships between stakeholders, and the capability to personalize STTs allow tourists to gain experiences that could not be obtained without such technologies [16]. Thus, the attributes of STTs can enhance tourists' ME [9,17]. Nonetheless, most of the studies in this area have been carried out in developed countries [6,18]. This is despite the fact that urban and rural areas in developing countries are also making rapid progress with regard to adapting themselves to the latest developments in smart technologies [3,8,16]. Indeed, citizens in these countries commonly use up-to-date smart technologies for their personal affairs [19]. There is also considerable evidence indicating the growing use of smart technologies and STTs in rural destinations located in developing countries after the COVID-19 outbreak [4]. In general, given the nature of rural regions, it seems that the experience of visiting smart rural destinations is different from the experiences gained through traveling to rural destinations on tours that follow traditional tourism procedures, or even smart urban destinations [6,20]. There is, however, little evidence to show to what extent the attributes of STTs can influence tourists' ME in rural areas [8,21–24]. Thus, the primary aim of the present study is to examine the effect of the attributes of STTs on tourists' ME in rural areas that have recently begun to employ smart technologies.

Users require special skills and competencies to interact with smart technologies and fulfill their needs [12,13,16]. Tourists' competence and knowledge of using STTs provide an opportunity for them to improve their ME [8,11]. In other words, tourists with sufficient knowledge and skills in using STTs are able to properly experience the full potential of these technologies while planning for their trip or during their visit [14,25]. Since such tourists can use the options offered by STTs, they are more likely to gain unique experiences in comparison with those of their counterparts who utilize such technologies in a conventional manner [12]. In fact, using STTs requires a knowledge base to facilitate the tourists' use of these technologies and the fulfillment of their needs [14,16,25]. Therefore, the second aim of this study was to explore the moderating role of user competence in the relationship between the attributes of STTs and tourists' ME.

Understanding the association between the tourists' use of STTs in the destinations and their experience during a visit is critical. This knowledge is key to enhancing visitor satisfaction and encouraging positive behavioral intention [26]. There is a wealth of evidence supporting the idea that gaining ME improves tourists' satisfaction and boosts their revisit intention [27]. Individuals who gain ME through exploiting the potential of STTs are more satisfied with their visit and are thus more likely to revisit the destination [21,28]. In fact, tourists' experiences when using STTs in tourism destinations are an integral part of their overall satisfaction with the destination, willingness to engage in WOM recommendations, and revisit intention [29–32]. Hence, the third purpose of this study was to investigate tourists' ME in using STTs and their impact on their satisfaction, willingness to engage in WOM recommendation, and intention to visit the destination again.

## 2. Theoretical Background and Hypotheses

During the COVID-19 pandemic, a large number of tourists turned to visiting destinations in rural areas [1]. However, many of these destinations enjoyed a lower level of smart technologies and could not adequately address the tourists' needs. Given the high demand for trips to rural areas and tourists' expectations of access to acceptable standards of smart services, ICT infrastructures began to develop in rural destinations [33,34]. In response to this demand, new smart rural destinations emerged that offered services to tourists. Yu et al. [26] believe that the pace of technological, economic, and social changes dictated by the pandemic has completely shifted the interactions between tourists and smart technologies in destinations. COVID-19 has changed the attitudes of many tourism stakeholders in relation to using smart technologies before and during their travels [35]. A large number of studies have concluded that tourists' dependence on smart technologies

has risen after the pandemic. As a result, managers and policymakers in such destinations should adapt their activities in light of stakeholders' new needs and expectations by improving the incorporation of smart technologies [36,37].

It appears that tourists can gain more ME and higher satisfaction through STTs according to their capabilities and skills [24]. Destinations offering digital services that are not found in other rural destinations appear more likely to encourage tourists to engage in WOM and pursue their intention to visit again [38]. However, there is no information about the impact of developing STTs in rural areas on tourists' ME and behavioral intentions. Attempts are made to address this research gap in the present study.

### 2.1. STTs

Smart destinations are characterized by a complex set of technologies (hardware and software infrastructure), individuals (creativity, variety, and training), and institutions (government and policy) [7,39,40]. Ling-Yun, Nao, and Min [41] believe that smart tourism destinations possess three essential components, namely, cloud services, the Internet of Things (IoT), and end-user internet service systems [42]. Cloud services provide access through internet websites, browsers, applications, and software programs [43]. The IoT makes it possible for smart destinations to observe and control the devices connected to the network by interacting with such devices and with those users with authorized access to the network [44]. The end-user internet service system provides access to payment methods, telecommunication interfaces, and wireless connections (e.g., hotspots and mobile data) for tourists via supporting devices and applications. STTs digitalize systems, processes, and services [7,45].

STTs entail all forms of online tourism applications and databases such as online travel agencies, personal weblogs, public websites, companies' websites, social media, smart telephones, and smart telephone applications, which can be utilized at any stage of their trip [16]. These technologies have four attributes, comprising accessibility, informativeness, interactivity, and personalization [9,46]. These attributes promote the usability and usefulness of STTs. Accessibility refers to the degree to which digital tourism resources are available and used by tourists and describes the ways in which tourists can gain access to information inside and outside the tourism destinations through STTs [16]. Many studies have indicated that accessibility is a vital factor in improving tourists' memorable experiences [22,47]. However, some studies in developed countries have shown that since STTs are pervasive, accessibility is not a determining factor for boosting tourists' memorable experiences [8]. Nevertheless, as rural regions in developing countries offer lower access to STTs, enhancing accessibility in these areas can improve tourists' experiences [9].

The informativeness of STTs is related to the reliability of digital tourism resources [22,23,48]. This attribute not only enhances tourists' knowledge of the destination [49] but also improves the speed of their access to general information during their trip, which, in turn, increases the quality of their visit [8,38]. For example, the COVID-19 pandemic heightened tourists' sensitivity regarding gaining high-quality and reliable information [9,50]. Concern over the virus in the context of destinations encouraged tourists to collect information using smart technologies [23]. Even after the outbreak began to slow down, tourists still needed to gain information about their destination, given the fear of pandemic resurgence [1,3]. It seems that information is even more important for tourists visiting rural areas because such destinations are geographically isolated [2,26]. However, they may suffer from some limitations regarding gaining speedy access to information [3,35]. As a result, it is hypothesized that improving the informativeness of STTs in rural areas enhances the tourists' peace of mind, thereby creating new and enjoyable experiences for them. The interactivity of STTs leads to a series of mutually beneficial relationships at the destination. In addition, the interactivity of STTs results in providing more relevant and reliable information, facilitated by the users' active participation [9,16]. This interactivity also makes it possible for tourists to get in touch with individuals outside the destination area and share their virtual content related to the rural destination. Interactivity inside and outside rural destinations has

always been a major challenge in developing countries [51]. As such, this attribute is likely to be attractive to tourists and may positively influence their experiences [21,52]. Finally, personalization refers to the tourists' ability to gain certain information in light of the requirements of their travel plans. It results in the fulfillment of individuals' needs within the shortest possible time [15,53]. Based on the relationships revealed in the literature, the following hypotheses are proposed:

**Hypothesis 1 (H1):** *The accessibility of STTs positively influences tourists' ME.*

**Hypothesis 2 (H2):** *The informativeness of STTs positively influences tourists' ME.*

**Hypothesis 3 (H3):** *The interactivity of STTs positively influences tourists' ME.*

**Hypothesis 4 (H4):** *The personalization of STTs positively influences tourists' ME.*

### 2.2. User Competence (Indirect Effects)

In order to fully leverage the potential of STTs, users must acquire the necessary knowledge and skills [13,19]. This implies that a certain level of user competence is essential to creatively and innovatively utilize all the attributes of smart technologies [12,16]. User competence in exploiting STTs can be defined as the capability of tourists to employ STTs in innovative and creative ways, enabling them to engage in unique activities in specific situations. Munro et al. [14] proposed three dimensions to characterize user competence: finesse, breadth of knowledge, and depth of knowledge.

The dimension of breadth of knowledge encompasses the user's proficiency in effectively employing a wide array of tools, skills, and knowledge to satisfy their needs [11]. It highlights their capacity to leverage diverse resources and capabilities associated with STTs to achieve desired outcomes. Users with a broad knowledge base can navigate through various features and functionalities, utilizing them to their advantage. Conversely, depth of knowledge reflects the degree to which users possess expertise in and mastery of a specific technology [54]. It pertains to their comprehensive understanding of the intricacies, functionalities, and underlying mechanisms of the technology. Users with a profound knowledge of a particular STT can maximize its potential and exploit its attributes to fulfill their specific requirements more effectively [25]. Finesse, the third dimension of user competence, denotes the user's ability to employ technologies in a creative and innovative manner. It captures their capacity to go beyond conventional usage patterns and explore novel ways of utilizing STTs [11,12]. Users with finesse can "think outside the box", adapting and customizing the technology to suit their unique needs and preferences. They possess a knack for innovative problem-solving and can extract the maximum value from the available features and functionalities [14].

Together, these dimensions of user competence—breadth of knowledge, depth of knowledge, and finesse—complement one another in enabling users to fully exploit the capabilities of STTs. By developing a broad understanding, profound expertise, and a creative mindset, users can harness the true potential of STTs and unlock memorable experiences on their travels [55].

The significance of user competence in utilizing STTs became particularly pronounced among tourists in light of the far-reaching effects of the COVID-19 pandemic [56]. With heightened concerns about health and safety, travelers who possessed advanced skills in using STTs were able to effectively reduce their physical contact with others [57]. By leveraging the capabilities of STTs, such as contactless transactions, virtual tours, and real-time information updates, these tourists could navigate their traveling experiences with greater confidence and minimize the potential risks associated with close interactions [56].

As a result, during the pandemic, many tourists actively sought to improve their competence in utilizing STTs [11,12]. They recognized the value and utility of these technologies in facilitating safer and more convenient traveling experiences [9]. By familiarizing themselves with the features, functionalities, and best practices of various STTs, tourists aimed

to enhance their ability to leverage these tools effectively. Whether it involved booking accommodation, accessing destination information, or engaging in virtual experiences, their improved competence in using STTs empowered them to navigate the challenges presented by the pandemic and make informed decisions to protect their well-being [12].

This heightened awareness and the pursuit of user competence in utilizing STTs during the COVID-19 pandemic underscored the importance of digital proficiency and adaptability in the realm of tourism [58]. It highlighted the role of technology in enabling travelers to overcome obstacles, ensure safety, and enhance their visit as a whole [9,56,59].

Overall, it is evident that user competence in utilizing STTs plays a pivotal role in influencing tourists' ME [11,19]. Consequently, in the present study, user competence in exploiting STTs was considered to be a mediating variable [12,16]. The assumption is that user competence mediates the relationship between the attributes of STTs and the tourists' ME [38]. Based on this assumption, the following hypotheses were formulated:

**Hypothesis 5 (H5):** *User competence mediates the relationship between the accessibility of STTs and tourists' ME.*

**Hypothesis 6 (H6):** *User competence mediates the relationship between the informativeness of STTs and tourists' ME.*

**Hypothesis 7 (H7):** *User competence mediates the relationship between the interactivity of STTs and tourists' ME.*

**Hypothesis** 8 **(H8):** *User competence mediates the relationship between the personalization of STTs and tourists' ME.*

*2.3. Memorable Experiences, Satisfaction, and Behavioral Intentions*

In this study, a memorable experience refers to a unique experience obtained via STTs that is highly regarded by tourists [9]. Smart tourism destinations try to improve tourists' ME by adopting smart tourism technologies and benefitting from their attributes [23]. Using STTs can make tourists aware of the depth and breadth of tourism activities [60,61]. Indeed, when tourists gain access to plenty of information and possible tourism activities, this immerses them in the tourism destination, leading to the creation of ME [8]. Digital services are less sophisticated in rural tourist destinations in developing countries [31,62]. Therefore, the provision of digital services is expected to offer unique experiences for tourists in such destinations [55].

Satisfaction refers to a tourist's positive assessment of their traveling experience [38,63]. When tourists are able to use STTs at different stages of their trip to make better decisions and register ME, they will record a higher level of satisfaction [16]. Many studies have shown that positive experiences during their visit enhance tourists' satisfaction [19,21]. Previous research suggests that satisfaction directly affects behavioral intention [9]. In fact, satisfaction mediates the relationship between experience and intention [8,31]. A tourist's intention to revisit a particular destination can depend on the quality of smart technology services provided in that destination [64]. A tourist's intention to return to a destination is itself memorable; an enjoyable visit motivates them to suggest the destination to others [27]. Many studies have indicated that satisfaction predicts tourists' revisit intention and their willingness to engage in WOM [38]. Higher levels of satisfaction positively influence tourists to return to a destination and to recommend it to others [16,31]. The quality of STT services provided in a specific destination plays an important role in boosting tourists' satisfaction, their intention to visit again, and their willingness to be involved in WOM [55]. However, the level of smart services provided in rural areas is not comparable with those offered in urban destinations [65–67]. Thus, if a minimum level of digital services is offered to tourists in rural destinations, they are more likely to engage in WOM to suggest that destination to others, and their revisit intention increases [27,29].

Overall, memorable experiences are an important factor in making future decisions, positively impacting satisfaction and their intention to visit the destination again [32]. Zhang, Sotiriadis, and Shen [55] showed that recording more ME during a visit enhances tourists' satisfaction and their revisit intention [8,22,31,68] Such tourists are also more likely to be involved in WOM to encourage others to visit the destination in question [64]. Given the development of smart technologies in rural areas in recent years, we assume that tourists are capable of using STTs on a wider scale and at a deeper level to register more ME [8,21,38]. Enhanced ME, in turn, increases tourists' satisfaction, their revisit intention, and their willingness to engage in WOM [69]. Based on the relationships delineated in the literature review, the following hypotheses are proposed:

**Hypothesis 9 (H9):** *Tourists' ME positively affect their satisfaction with rural tourism destinations.*

**Hypothesis 10 (H10):** *Tourists' satisfaction positively affects their intention to revisit rural tourism destinations.*

**Hypothesis 11 (H11):** *Tourists' satisfaction positively affects their willingness to be engaged in WOM regarding rural tourism destinations.*

The proposed model describes how tourists' ME, their revisit intention, and their willingness to engage in WOM are boosted through STTs (Figure 1). This research model was developed at three levels. The first level concerns H1, H2, H3, and H4. This level deals with the impact of the attributes of STTs (i.e., accessibility, interactivity, informativeness, and personalization) on tourists' ME. The second level, which addresses H5, H6, H7, and H8, explores the mediating role of user competence. At the third level, H9, H10, and H11 are assessed by examining the role of ME in raising tourists' levels of satisfaction and intention to visit the destination again. This level also explores the effect of tourists' satisfaction on their revisit intention. Based on the above-proposed hypotheses, the conceptual framework of this study is developed. This framework is presented in Figure 1, below.

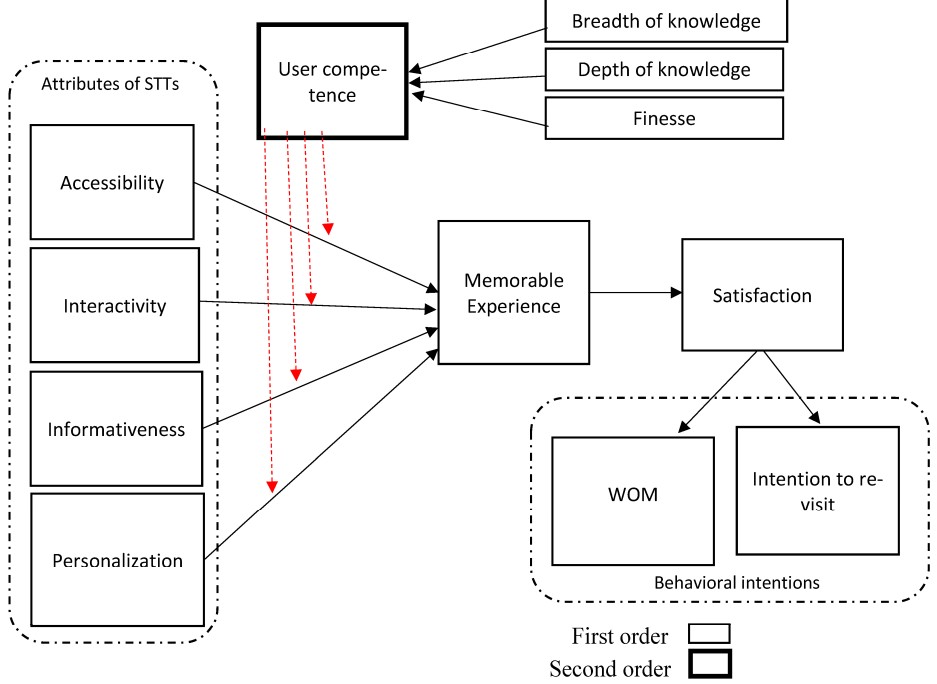

**Figure 1.** Conceptual framework of the study.

### 3. Background

To assess the proposed model, five villages in Shemiranat County in Iran, located to the north of Tehran, were selected for data collection. These villages succeeded in enhancing their use of smart technologies during the pandemic and are now popular tourist destinations [56,70,71]. During the COVID-19 pandemic, it was essential to use smart technologies to observe social distancing and fulfill citizens' and tourists' daily needs [1,72]. Although, at the time, the revenues of Iran's government and private sector had significantly dwindled as a result of the imposed sanctions, they could still direct their policies toward developing smart technologies in rural areas [73,74]. Based on official reports, over the past two years, huge investment has been made into developing an ICT infrastructure and executing capacity-building programs to boost people's awareness and skills in using smart technologies [75]. It seems that the development of ICT and the improvement of digital literacy have profoundly impacted tourism, which is an important part of the rural economy [56,76]. Thus, new rural destinations have emerged due to the increasing use of STTs.

The selected villages for this study were chosen based on criteria such as being national heritage sites with historic construction, unique architectural styles, and attractive natural landmarks [56]. These villages possess a rich historical background and offer significant cultural and natural assets; they are also situated in regions with favorable mountainous climates. Annually, they attract a considerable number of domestic and international tourists, with a wide range of tourism tours available. Their proximity to Tehran has contributed to the development of these villages, enabling them to benefit from its excellent infrastructure and attract a growing number of visitors [74]. Notable examples of such villages in Shemiranat County include Afjeh, Ahar, Feshm, Sink, Abnik, Berg-e Jahān, Naran, and others. These villages offer ample welfare and sanitary facilities, as well as extensive and suitable public and tourist infrastructures [56]. With their rich historical background, they provide appealing destinations for culture- and nature-oriented tourists who wish to enjoy their leisure time. The locations of these villages can be identified on Map 1.

### 4. Methodology

A quantitative research design was adopted to assess the proposed model. The researchers developed measurement scales to assess two of the variables in light of previous studies and according to the objectives of the current study (Table 1). The study encompasses nine constructs, with eight of them (i.e., accessibility, personalization, interactivity, informativeness, ME, satisfaction, revisit intention, and willingness to engage in WOM) being reflective. The last variable (i.e., user competence) is a second-order reflective-formative construct with three dimensions comprising finesse, breadth of knowledge, and depth of knowledge. These three dimensions are reflective and formatively establish user competence.

**Table 1.** Variables and measures used in this study to test the research hypotheses.

| Variable | Measure | References |
|---|---|---|
| Accessibility | In this village, I was able to access the potential of smart technologies anytime and anywhere. | [9,19,22,47,55,77,78] |
| | In this village, smart tourism technologies (e.g., Wi-Fi, smart telephone services, smart technologies in accommodation, etc.) were easily accessible. | |
| | In this village, I could easily gain access to smart tourism technologies via other relevant websites. | |

**Table 1.** *Cont.*

| Variable | Measure | References |
|---|---|---|
| Informativeness | In this village, smart tourism technologies offered useful information about my visit. | [9,19,22,23,38,48] |
| | Smart tourism technologies helped me to choose this village as my destination. | |
| | Smart tourism technologies helped me to gain information about the outside world rapidly. | |
| | During my trip, smart tourism technologies, tourism applications, and websites fulfilled my needs. | |
| Interactivity | Smart tourism technologies helped me to engage in Q & A during my visit, share my comments, and interact with others. | [8,9,16,19,31,38,79] |
| | During the trip, I was able to share information about rural tourism technologies easily using smart tourism technologies. | |
| | During the trip, I could easily interact with others via smart tourism technologies. | |
| Personalization | Smart tourism technologies made it possible for me to access suitable information. | [8,19,22,38] |
| | Smart tourism technologies offered the best choices of routes and information during the trip in light of the conditions. | |
| | Smart tourism technologies made it possible for me to personalize information searches about tourism and tourism destinations. | |
| | The information that was personalized through smart tourism technologies addressed my needs related to tourism and travel. | |
| Breadth of knowledge | I have enough knowledge to use smart tourism technologies. | [11–14,25,80–82] |
| | I have sufficient skills to use smart tourism technologies. | |
| | I cannot use smart tourism technologies without the help of others. | |
| Depth of knowledge | I have enough knowledge to install new applications related to smart tourism technologies on my cellphone. | [11–14,25,80–82] |
| | I know how to connect smart tourism technologies with the internet. | |
| | During my travels, I use smart options such as the camera, video recording, and voice recording via smart tourism technologies. | |
| Finesse | I use smart technologies to solve my problems. | [11–14,25,80–82] |
| | I feel that I have enough creativity to use smart technologies to conduct my affairs. | |
| | I am innovative in using smart technologies. | |
| WOM | Given the available smart tourism technologies, I will recommend this village to my family and friends. | [38,49,83,84] |
| | Given the available smart tourism technologies, I will recommend this village to others. | |
| | I will share the positive points about the smart tourism technologies of this village with others. | |
| Memorable experiences | The applications of smart technology in this village recorded memorable experiences for me. | [8,9,24,55,85] |
| | The applications of smart technology helped me have a good trip by registering unique experiences. | |
| | Using the applications of smart technology helped me have a more memorable visit. | |

**Table 1.** *Cont.*

| Variable | Measure | References |
|---|---|---|
| Satisfaction | The quality of smart tourism technologies in this village was satisfactory for me. | [37,55,60,86,87] |
| | Smart tourism technologies in this village were beyond my expectations and made my visit joyful. | |
| | In gaining unique experiences, using smart tourism technologies was ideal and thrilled me. | |
| Revisit intention | Given its smart technologies, I will visit this village again in the future. | [22,24,26,31,37,55] |
| | I will recommend this village to my family and friends because of its smart technology capabilities. | |
| | I will share the memorable experiences that I gained by using smart tourism technologies with others. | |

*Data Collection Procedures*

The data collection for this study took place between April and July 2022, utilizing a convenience sampling method. The target population consisted of tourists visiting five emerging destinations located in Shemiranat County. To ensure the accuracy and reliability of the collected data, a team of trained research assistants was recruited and actively engaged in the data collection process. Recognizing that the term STT may be unfamiliar to some tourists, the research assistants provided a comprehensive explanation of the concept and research objectives for the respondents.

Before initiating the data collection process, the research team conducted an extensive analysis of statistical data obtained from the Ministry of Tourism of Iran. This analysis revealed a significant growth in the utilization of smart tourism technologies in the targeted rural areas in recent years. Furthermore, preliminary qualitative research conducted by the team confirmed the proliferation of smart technologies in these regions. Armed with this knowledge, the research team focused their data collection efforts on individuals who had traveled to these rural destinations following the COVID-19 pandemic, with the majority of the participants originating from Tehran, the capital of Iran [56].

To gather the required information, the research team strategically conducted data collection activities in various locations within these rural areas, including rural residences and recreation centers. These locations were chosen to ensure a representative sample of tourists who had experienced the smart technologies available in these villages. By gathering data from these specific locations, the research team aimed to capture the perspectives and experiences of tourists who had firsthand exposure to the smart tourism technologies offered in these rural settings.

During the two-month data collection period, a total of 630 surveys were distributed to the targeted tourists. Following the completion of the survey, 590 questionnaires were returned and were deemed suitable for analysis, ensuring a robust dataset for conducting the subsequent research analysis. To ensure the normality of the data, the research team employed the Kolmogorov–Smirnov test, which is a widely used statistical test for assessing the distribution of data. The results of the test indicated that the data collected from the surveys were within the normal range, satisfying the assumption of multivariate normality in the subsequent analysis.

Overall, 55.25% of the respondents were male, and 44.75% were female; 49.4% of them had already visited the village and the rest (50.6%) were first-time visitors. Moreover, 26.94% of the participants had utilized STTs prior to their visit, 73.05% had used them to plan their travel, and 97.45% used such technologies during their visit. Around 39.5% of the respondents identified themselves as Generation Y, 21.8% as Generation X, and 38.7% as Generation Z (Table 2).

**Table 2.** Demographic Profile.

| Characteristics | | Frequency% | Characteristics | | Frequency% |
|---|---|---|---|---|---|
| Gender | Female | 264 (44.75) | Education | Did not complete high school | 41 (6.94) |
| | | | | High school degree or equivalent | 118 (20) |
| | Male | 326 (55.25) | | Bachelor | 324 (54.91) |
| | | | | Associate | 59 (10) |
| | | | | Graduate | 48 (8.15) |
| Generation | Y | 150 (39.5) | Smart devices used at the destination | Smartphone | 572 (97) |
| | X | 83 (21.8) | | Smart watch | 100 (17) |
| | Z | 147 (38.7) | | Tablet | 88 (14) |
| Stage of the trip when using STTs | Before a trip | 159 (26.94) | | Other | 325 (55) |
| | While planning | 431 (73.05) | | | |
| | While traveling | 575 (97.45) | | | |

## 5. Results

### 5.1. Measurement Models

An assessment of the measurement model involves evaluating the reliability and validity of the reflective constructs and establishing second-order composite constructs, based on the obtained values for different dimensions of user competence (see Table 3).

Initially, we examined the reliability and convergent validity of reflective constructs through factor loadings, Cronbach's alpha values, and composite reliability [88]. All items demonstrated loadings above the threshold of 0.4, indicating their suitability. Moreover, Cronbach's alpha values exceeding 0.70 and composite reliability values above 0.5 confirmed the reliability of all variables, ensuring satisfactory internal consistency in the measurement model. Divergent validity, a critical criterion in the partial least squares approach, was assessed by comparing the AVE values [89]. The measurement items exhibited higher loadings on their respective constructs and the square root of each construct's AVE exceeded its correlation coefficients, thus confirming discriminant validity [88].

In the second stage, we evaluated the second-order reflective–formative (composite) constructs, such as user competence, by examining the research's multi-collinearity using variance inflation factors (VIF) [69]. Table 3 presents the results, indicating VIF values below 5 and significant outer weights for the associated items of the user competence construct. To establish the discriminant validity of the formative and reflective constructs in the second stage, we assessed the full collinearity VIF, aiming for values below 3.3 [47]. The results showed that all constructs in the second stage had full collinearity VIF values lower than 3.3, confirming an acceptable level of discriminant validity [89].

The assessment involves comparing the average root of the AVE variables with the correlations of latent variables (LV). It is expected that the square root of each AVE variable surpasses its highest correlation with any other variable. Table 4 highlights the square roots of the AVE values in bold (0.955, 0.772, 0.771, 0.905, 0.884, 0.830, 0.781, 0.792, 0.795, 0.912, and 0.756). These values confirm the presence of discriminant validity, as they satisfy the necessary criterion.

**Table 3.** Results of the measurement model.

| Measure | | Loadings/Weights |
|---|---|---|
| Accessibility (CR = 0.797; Cronbach $\alpha$ = 0.750; AVE = 0.515) | | |
| In this village, I was able to access the potential of smart technologies anytime and anywhere. | Reflective | 0.908 |
| In this village, smart tourism technologies (e.g., Wi-Fi, smart telephone services, smart technologies in accommodation, etc.) were easily accessible. | | 0.998 |
| In this village, I could easily gain access to smart tourism technologies via other relevant websites. | | 0.958 |
| Informativeness (CR = 0.846; CA = 0.754; AVE = 0.587) | | |
| In this village, smart tourism technologies offered useful information about my visit. | Reflective | 0.865 |
| Smart tourism technologies helped me to choose this village as my destination. | | 0.701 |
| Smart tourism technologies helped me gain information about the outside world rapidly. | | 0.832 |
| During my visit, smart tourism technologies, tourism applications, and websites fulfilled my needs. | | 0.705 |
| Interactivity (CR = 0.923; CA = 0.884; AVE = 0.756) | | |
| Smart tourism technologies helped me to engage in Q & A, share my comments, and interact with others during my visit. | Reflective | 0.849 |
| During my visit, I was able to easily share information about rural tourism technologies using smart tourism technologies. | | 0.952 |
| During my visit, I could easily interact with others via smart tourism technologies. | | 0.889 |
| Personalization (CR = 0.903; CA = 0.839; AVE = 0.757) | | |
| Smart tourism technologies made it possible for me to access useful information. | Reflective | 0.750 |
| Smart tourism technologies recommended the best routes and information during the trip in light of the circumstances. | | 0.748 |
| Smart tourism technologies made it possible for me to personalize information about tourism and tourist destinations. | | 0.842 |
| The information personalized through smart tourism technologies addressed my needs related to tourism and travel. | | 0.827 |

**Table 3.** *Cont.*

| Measure | | Loadings/Weights |
|---|---|---|
| **WOM (CR = 0.821; CA = 0.701; AVE = 0.610)** | | |
| Given the available smart tourism technologies, I will recommend this village to my family and friends. | | 0.946 |
| Given the available smart tourism technologies, I will recommend this village to others. | Reflective | 0.702 |
| I will share the positive points about the smart tourism technologies of this village with others. | | 0.717 |
| **Memorable experiences (CR = 0.989; CA = 0.831; AVE = 0.746)** | | |
| The application of smart technology in this village recorded memorable experiences for me. | | 0.972 |
| The application of smart technology helped me to have a good trip by registering unique experiences. | Reflective | 0.818 |
| Using the application of smart technology helped me have a more memorable trip. | | 0.919 |
| **Satisfaction (CR = 0.868; CA = 0.771; AVE = 0.687)** | | |
| The quality of smart tourism technologies in this village was satisfactory for me. | | 0.861 |
| Smart tourism technologies in this village were beyond my expectations and made my visit joyful. | Reflective | 0.881 |
| Gaining unique experiences through using smart tourism technologies was ideal and thrilled me. | | 0.742 |
| **Revisit intention (CR = 0.880; CA = 0.791; AVE = 0.712)** | | |
| Given its smart technologies, I will visit this village again in the future. | | 0.719 |
| I will recommend this village to my family and friends because of its smart technologies. | Reflective | 0.907 |
| I will share the memorable experiences gained by using smart tourism technologies with others. | | 0.894 |
| **Breadth of knowledge (CR = 0.880; CA = 0.791; AVE = 0.712)** | | |
| I have enough knowledge to use smart tourism technologies. | | 0.875 |
| I have sufficient skills to use smart tourism technologies. | Reflective | 0.747 |
| I cannot use smart tourism technologies without the help of others. | | 0.756 |

**Table 3.** *Cont.*

| Measure | | Loadings/Weights | | |
|---|---|---|---|---|
| Depth of knowledge (CR = 0.880; CA = 0.791; AVE = 0.712) | | | | |
| I have enough knowledge on how to install new applications related to smart tourism technologies on my cellphone. | Reflective | 0.850 | | |
| I know how to connect smart tourism technologies with the internet. | | 0.747 | | |
| During the trip, I use smart options such as the camera, video recording, and voice recording capabilities of smart tourism technologies. | | 0.756 | | |
| Finesse (CR = 0.880; CA = 0.791; AVE = 0.712) | | | | |
| I use smart technologies to solve my problems. | | 0.757 | | |
| I feel I have enough creativity to use smart technologies to conduct my affairs. | Reflective | 0.714 | | |
| I am innovative in using smart technologies. | | 0.840 | | |
| | | Composite | *p*-value | VIF |
| User competence | Breadth of knowledge | 0.271 | <0.000 | 2.054 |
| | Depth of knowledge | Composite | 0.242 | <0.000 | 1.86 |
| | Finesse | 0.230 | <0.000 | 1.745 |

Note: AVE = average variance extracted. CR = composite reliability. CA = Cronbach's alpha.

**Table 4.** Discriminant validity.

| | A | F | INF | ME | RI | S | WOM | BOK | DOK | INT | P |
|---|---|---|---|---|---|---|---|---|---|---|---|
| A | 0.955 | | | | | | | | | | |
| F | 0.742 | 0.772 | | | | | | | | | |
| INF | 0.769 | 0.614 | 0.771 | | | | | | | | |
| ME | 0.860 | 0.776 | 0.701 | 0.905 | | | | | | | |
| RI | 0.550 | 0.534 | 0.417 | 0.521 | 0.844 | | | | | | |
| S | 0.789 | 0.670 | 0.712 | 0.823 | 0.443 | 0.830 | | | | | |
| WOM | 0.740 | 0.694 | 0.773 | 0.864 | 0.645 | 0.723 | 0.781 | | | | |
| BOK | 0.759 | 0.766 | 0.626 | 0.811 | 0.528 | 0.695 | 0.698 | 0.792 | | | |
| DOK | 0.604 | 0.528 | 0.443 | 0.564 | 0.703 | 0.467 | 0.732 | 0.540 | 0.795 | | |
| INT | 0.779 | 0.766 | 0.714 | 0.857 | 0.403 | 0.808 | 0.703 | 0.763 | 0.426 | 0.912 | |
| P | 0.711 | 0.610 | 0.555 | 0.711 | 0.544 | 0.735 | 0.704 | 0.645 | 0.564 | 0.604 | 0.756 |

Note: A = Accessibility, F = finesse, INF = informativeness, ME = memorable experience, RI = revisit intention, S = satisfaction, WOM = word of mouth, BOK = breadth of knowledge, DOK = depth of knowledge, INT = interactivity, P = personalization.

### 5.2. Testing the Hypotheses

Table 5 shows the results of testing the hypotheses in this study. Accordingly, accessibility has a significant positive effect on ME ($\beta = 0.551$, $p < 0.000$, T = 12.159); hence, H1 is confirmed. Informativeness also exerts a significant positive influence on ME ($\beta = 0.230$, $p < 0.000$, T = 15.599), meaning that H2 is supported. Moreover, interactivity also exerts a significant positive influence on ME ($\beta = 0.052$, $p < 0.005$, T = 2.844), meaning that H3 is supported. However, personalization has no influence on ME ($\beta = 0.079$, $p < 0.238$, T = 1.180), meaning that H4 is not supported.

**Table 5.** Path coefficient and hypothesis testing (direct influence and mediation).

| Hypothesis | (β) | T-Value | *p*-Value | Supported |
|---|---|---|---|---|
| H1:accessibility → (+) Memorable Experience | 0.551 | 12.159 | 0.000 | (Direct influence) YES |
| H2: Informativeness → (+) Memorable Experience | 0.230 ** | 15.599 | 0.000 | (Direct influence) YES |
| H3: interactivity → (+) Memorable Experience | 0.052 ** | 2.844 | 0.005 | (Direct influence) YES |
| H4: personalization → (+) Memorable Experience | 0.079 | 1.180 | 0.238 | (Direct influence) NO |
| H5: accessibility → user competence → (+) Memorable Experience | 0.057 ** | 2.900 | 0.002 | (Mediation) Partial |
| H6: Informativeness → user competence → (+) Memorable Experience | 0.051 ** | 2.860 | 0.001 | (Mediation) Partial |
| H7: interactivity → user competence → (+) Memorable Experience | 0.038 ** | 2.872 | 0.004 | (Mediation) Partial |
| H8: personalization → user competence → (+) Memorable Experience | −0.010 | 0.693 | 0.489 | (Mediation) NO |
| H9: Memorable Experience → (+) Satisfaction | 0.823 ** | 72.556 | 0.000 | (Direct influence) YES |
| H10: Satisfaction → (+) Revisit intention | 0.443 ** | 12.474 | 0.000 | (Direct influence) YES |
| H11: Satisfaction → (+) WOM | 0.723 ** | 47.210 | 0.000 | (Direct influence) YES |

Note: ** $p < 0.05$.

Tourists' competence in using STTs partially mediates the relationship between accessibility and ME (β = 0.057, $p < 0.002$, T = 2.900), meaning that H5 is supported. It also partially mediates the relationship between informativeness (β = 0.051, $p < 0.001$, T = 2.860) and interactivity (β = 0.038, $p < 0.004$, T = 2.872), on the one hand, and ME, on the other hand, thus confirming H6 and H7. However, user competence is not a significant variable mediating the relationship between personalization and ME (β = −0.010, $p < 0.693$, T = 0.489). Thus, H8 is not supported. In total, accessibility, interactivity, informativeness, user competence, and personalization account for 97% of the variance in ME.

The results also provide evidence supporting H9, indicating a significant positive effect of ME on tourists' satisfaction (β = 0.823, $p < 0.000$, T = 72.556). Furthermore, tourists' satisfaction significantly influences their revisit intention (β = 0.443, $p < 0.000$, T = 12.474) and their willingness to engage in WOM (β = 0.723, $p < 0.000$, T = 47.210), confirming H10 and H11. Figure 2 illustrates that tourists' ME explains 67.7% of the variance in their satisfaction. Importantly, satisfaction accounts for 19.5% of the variance in revisit intention and 52.2% of the variance in WOM, as depicted in Figure 2.

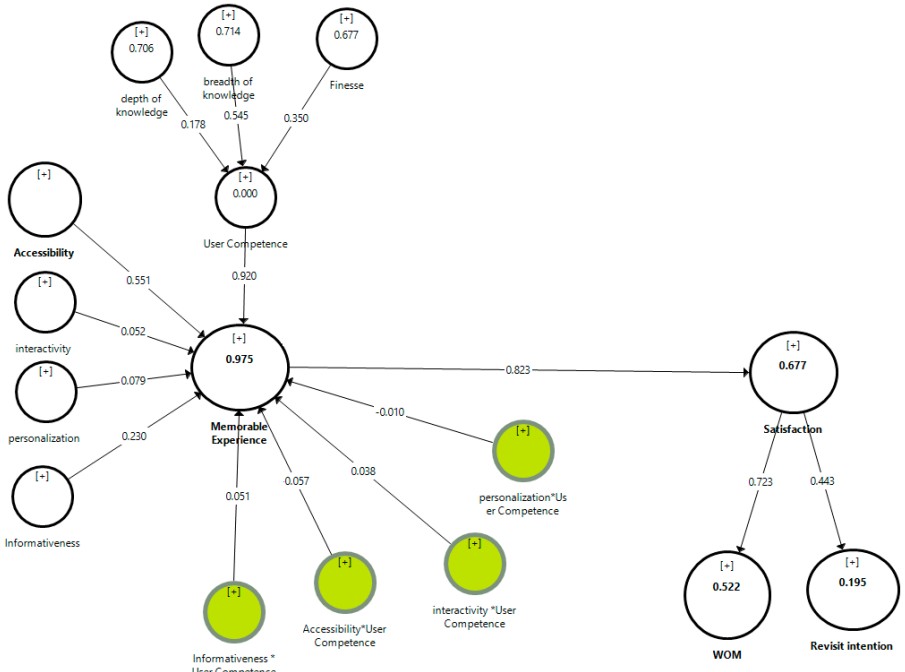

**Figure 2.** Results of the structural model.

## 6. Discussion and Conclusions

The results of this study showed that tourists widely used STTs in emerging rural destinations to gain unique and memorable experiences ($R^2 = 0.975$), which, in turn, enhanced their satisfaction ($R^2 = 0.677$), their willingness to engage in WOM ($R^2 = 0.522$), and their intention to visit the destination again ($R^2 = 0.195$). Focusing on four key attributes of STTs, the current study tried to examine their effect on tourists' ME. The results showed that the accessibility of STTs comprises the main factor in maximizing tourists' ME in rural destinations. This may be related to the weak digital infrastructure of rural destinations in Iran [73,76]. Since a lack of sufficient accessibility is a major issue in rural areas in developing countries [90], offering good accessibility in these destinations boosts tourists' ME. In other words, tourists can gain ME in emerging rural destinations that they cannot accumulate in other rural areas. It seems that access to smart technologies is attractive for tourists in rural areas. However, Jeong and Shin [8] showed that accessibility is not a determining factor for tourists. This could be attributed to the fact that all areas of the United States enjoy a certain level of accessibility; hence, it is not a distinguishing factor in improving tourists' ME.

The informativeness of STTs is another factor exerting a profound effect on improving tourists' ME. It appears that this attribute helps tourists to collect valuable information about rural destinations, e.g., tourist attractions, weather, etc. It also helps them gain updates from outside the rural destination about important matters such as COVID-19-related news, bank account information, etc. Given that rural areas suffer from geographical isolation, being informed of the latest developments outside the destination via reliable sources can boost tourists' ME. The available evidence suggests that visiting rural areas typically reduces tourists' access to updates from outside the destination [2,91]. Smart rural destinations, however, allow tourists to easily access reliable information about the outside world, similar to urban regions. The results of this study further showed that interactivity has little effect on tourists' ME. This attribute of STTs enhances the mutual interaction between tourists and their friends, acquaintances, and service providers. It seems that sharing attractions and experiences with friends and acquaintances helps tourists to record higher levels of ME. Indeed, the interactivity of STTs in rural areas allows tourists to interact with individuals and groups and to share events and experiences in the same way as they can in urban regions. As a result, this attribute leads to recording unique and memorable travel experiences.

Moreover, the interaction between tourists and service providers in the destination improves the quality of offered services and tourists' contribution. The results indicated that personalization did not significantly affect tourists' ME. In contrast to this finding, Huang et al. [16] demonstrated that personalization is an important variable in registering ME. Due to the lack of sufficient STT sophistication in rural areas, personalization is not of paramount importance.

Venkatesh, Thong, and Xu [92] demonstrated that competence is not an important factor in determining tourists' behavior. The results of the current study showed that tourists' competence in using STTs in emerging rural destinations partially mediates the relationships between the three STT attributes (i.e., informativeness, interactivity, and accessibility) and tourists' memorable experiences. In other words, the more competent the user, the more memorable the experience that they will record. Many studies have also indicated that possessing knowledge, skill, and creativity when using STTs for planning their travel and during the visit can lead to unique experiences for tourists [11,12,19,25].

The results showed that people who register unique experiences through STTs are more satisfied with their choice of destination. Gaining ME while at the destination enhances tourists' satisfaction. The results showed that using STTs boosts tourists' intention to return and their willingness to engage in WOM in rural areas. Their satisfaction with the smart technologies offered in rural areas encourages tourists to vouch for these tourist destinations [27,55].

### 6.1. Theoretical Implications

This study makes significant contributions to the theoretical understanding of smart tourism, addressing several gaps in the existing literature. The findings of this research have important theoretical implications that enrich the field in the following ways. First, the study focuses on the factors influencing tourists' revisit intention and their willingness to engage in WOM in emerging and smart destinations, following the spread of COVID-19. This is a topic that has received limited attention in the tourism literature. The results demonstrate that the attributes of STTs enhance tourists' revisit intention and their propensity to engage in WOM in rural destinations. As there are few smart rural destinations in developing countries, the study reveals that tourists are inclined to revisit such destinations and share their unique experiences, facilitated by the attributes of STTs via WOM. This expands our understanding of the role of STTs in enhancing tourist engagement and their loyalty to emerging destinations.

Second, the study breaks new ground by focusing on smart rural destinations in a developing country. Unlike previous studies that predominantly concentrated on urban areas and developed countries, this research shifts the spotlight to rural contexts. The investigation identifies the accessibility of STTs as a crucial factor in improving ME. This novel perspective offers valuable insights into the specific needs and preferences of tourists in emerging and smart rural destinations, contributing to a more comprehensive understanding of the impact of STTs in diverse tourism contexts [8,16,21,44].

Third, the study addresses a significant research gap by highlighting tourists' competence in using smart technologies as a key factor in enhancing ME. While previous studies have primarily examined the direct relationship between the attributes of smart tourism and ME, this research reveals the mediating role played by user competence. The findings emphasize that tourists with higher skills and greater knowledge of using STTs are more likely to achieve enhanced ME. This innovative approach advances our understanding of the complex interplay between tourists' competencies and the utilization of STTs, enhancing our knowledge of the factors that influence ME in the context of smart tourism [8,16,21,44]. By shedding light on these underexplored aspects, this study significantly contributes to the theoretical foundations of smart tourism. It expands the literature by exploring the specificities of emerging and smart rural destinations, uncovering the role of user competence, and offering insights into tourists' intentions to return to a destination and their engagement through WOM. These theoretical implications enhance our understanding of the factors influencing tourists' experiences and their behaviors in the context of smart tourism.

### 6.2. Implications for Policy and Practice

The findings of this study have significant implications for the development and implementation of STTs, as well as for enhancing tourists' intention to revisit and their willingness to engage in WOM. To create favorable conditions for the effective use of STTs in rural destinations, it is essential for rural tourism managers in emerging areas to prioritize meeting the minimum performance standards in key attributes such as accessibility, interactivity, and informativeness. Given that tourists' primary objective is to maximize their use of smart technologies in rural areas to enhance their visit, they expect to receive reliable, useful, and easily accessible information. For example, during situations such as the COVID-19 pandemic, tourists should be able to utilize STTs to access fast and up-to-date information online. The adoption of new technologies can greatly enhance the attributes of STTs in rural destinations, leading to increased tourist satisfaction and overall destination attractiveness. Thus, managers and policymakers should actively explore and embrace new smart technologies, thereby adapting to tourists' evolving expectations and needs [19].

Furthermore, this study highlights the importance of investing in an ICT infrastructure and improving tourists' access to smart services in rural destinations. While the studied villages enjoyed a higher level of digitalization compared to other rural areas in Iran, additional public and private investments are necessary to further enhance tourists' access to smart services. Managers and policymakers should prioritize significant investments in

developing the ICT infrastructure for smart tourism in rural areas [40,47,57,93]. This will not only increase the tourists' revisit intention and their willingness to engage in WOM but will also contribute to the overall growth and development of rural destinations. The study also emphasizes the role of tourists' competence when using STTs to create ME. Marketers promoting emerging destinations are encouraged to provide opportunities and favorable conditions for tourists to improve their skills at various levels, including finesse, breadth of knowledge, and depth of knowledge, in order to enhance their ability to record more ME. In addition to providing the necessary infrastructure for the development of smart tourism and STTs in rural destinations, managers should employ direct and indirect methods to enhance tourists' knowledge and skills in utilizing STTs effectively. Offering STT training before and during travel, in collaboration with tour leaders or via interactive platforms, can significantly enhance the tourists' digital knowledge and skills. Marketers should also focus on attracting tourists who have a keen interest in smart technologies and who possess the necessary skills to utilize them.

Moreover, this study underscores the potential of rural destinations in developing countries to leverage the development of smart technologies as a means to attract tourists. These destinations, which, historically, have had limited access to smart technologies, can now offer relatively advanced digital services. This novelty can be an attractive feature for tourists who are not accustomed to having access to such services in rural settings [94]. Therefore, taking practical measures to develop and offer high-quality digital services in rural destinations can lead to their further development. This, in turn, is likely to increase the tourists' willingness to revisit these destinations and will encourage newcomers to explore and experience these rural areas. The practical implications of this study highlight the importance of proactive engagement with smart technologies, investment in ICT infrastructure, and the improvement of tourists' competence in utilizing STTs. By implementing these measures, rural destinations can enhance their attractiveness, increase tourists' satisfaction, and foster sustainable growth in rural tourism.

### 6.3. Limitations and Suggestions for Future Research

Despite the valuable insights gained from this study, there are some limitations that should be acknowledged, providing opportunities for future research endeavors. Firstly, it is important to note that the current study focused on collecting data from only six emerging smart destinations that are in close proximity to the megacity of Tehran. As these smart rural tourism destinations are closely intertwined with Tehran, the findings may differ when considering rural destinations that are near smaller cities. To enhance the generalizability of the findings, future research could explore other smart rural tourism destinations that are located in different world regions.

Furthermore, this study concentrated on a single developing country, potentially limiting the broader understanding of the impact of STTs on tourists' memorable experiences. Conducting similar research in other countries would enable comparisons and shed light on the potential similarities and differences in the effects of STTs on tourists' experiences.

Another aspect worth considering is the adoption of a cross-sectional design for data collection in this study. Given the rapid pace at which STTs are evolving, employing longitudinal designs in future studies could provide a more comprehensive understanding of the long-term effects of STTs in rural destinations.

Additionally, the finding that the accessibility of STTs had the strongest effect on tourists' memorable experiences in emerging rural destinations contradicted previous findings. Therefore, further research could delve into qualitative investigations to gain deeper insights into this association and explore the underlying factors contributing to this phenomenon. It is also important to acknowledge that this study did not specifically address the potentially negative effects of STTs on tourists' experiences, such as privacy concerns or an over-reliance on technology. Future research could consider incorporating these aspects to provide a more comprehensive understanding of the overall impact of STTs on tourists' experiences.

**Author Contributions:** Software, Z.-A.T. and M.P.; Validation, A.S. and F.J.; Formal analysis, Z.-A.T. and C.M.H.; Investigation, C.M.H.; Resources, Z.-A.T., M.P., C.M.H., A.S. and F.J.; Data curation, M.P.; Writing—original draft, M.P.; Writing—review & editing, A.S.; Supervision, Z.-A.T. All authors have read and agreed to the published version of the manuscript.

**Funding:** This research received no external funding.

**Conflicts of Interest:** The authors declare no conflict of interest.

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
